# Effects of Fungi on Soil Organic Carbon and Soil Enzyme Activity under Agricultural and Pasture Land of Eastern Türkiye

Erhan Erdel [1,*], Uğur Şimşek [1] and Tuba Genç Kesimci [2]

1 Department of Soil Science and Plant Nutrition, Faculty of Agriculture, Igdır University, Igdır 76000, Türkiye
2 Department of Plant Protection, Faculty of Agriculture, Igdır University, Igdır 76000, Türkiye
* Correspondence: erhan.erdel@igdir.edu.tr

**Abstract:** Soil organic matter (SOM) is a heterogeneous mixture of materials ranging from fresh plant tissues to highly decomposed humus at different stages of decomposition. Soil organic carbon (SOC) status is directly related to the amount of organic matter in soil and therefore is generally used to measure it. Soil carbon sequestration refers to the removal of carbon (C) containing substances from the atmosphere and its storage in soil C pools. The soil microbial community (SMC) plays an important role in the C cycle, and its activity is considered to be the main driver of differences in C storage potential in soil. The composition of SMC is crucial for maintaining soil ecosystem services, as the structure and activity of SMC also regulate the turnover and distribution of nutrients, as well as the rate of soil organic matter (SOM) decomposition. Here, we applied fungi on the soils taken from two fields, one used as a pasture and one for agriculture (wheat cultivation), in a histosol in the eastern part of Türkiye and investigated the changes in the organic carbon and enzyme activity contents of the soils at the end of the 41-day incubation period. In the study, four different fungal species (*Verticillium dahliae* (SOR-8), *Rhizoctonia solani* (S-TR-6), *Fusarium oxysporum* (HMK2-6), and *Trichoderma* sp.) were used and catalase, urease, and alkaline phosphatase activities were examined. Results showed that the values of SOC were *V. dahliae* (7.46%), *Trichoderma* sp. (7.27%), *R. solani* (7.03%), Control (6.97%), and *F. oxysporum* (6.7%) in pastureland and were *V. dahlia* (4.72%), control (4.69%), *F. oxysporum* (4.65%), *R. solani* (4.37%), and *Trichoderma* sp. (4.14%) in agricultural land, respectively. SOC and soil enzyme activities were significantly affected by land use types ($p < 0.05$). The higher SOC and enzyme activities were observed in pastureland. Finally, it was determined that soil organic carbon and soil enzyme activities were affected by fungi. This study is important in terms of revealing that the effects of fungi on soil organic carbon and enzyme activities are different in various land types.

**Keywords:** land use type; fungi; soil organic carbon; catalase activity; urease activity; alkaline phosphatase activity

## 1. Introduction

Soil carbon includes both soil organic matter and inorganic carbon (carbonate minerals). It is reported that the carbon stored in soil is higher than the carbon pool in the atmosphere [1,2] and is also the largest terrestrial carbon stock. A small change in this soil carbon pool has a major impact on the global climate [3] because soil organic carbon (SOC) plays a crucial role in the global carbon cycle. In addition, it has a significant effect on many processes taking place in soil. Improper soil management and the conversion of forest and pasturelands into fields can cause large losses of carbon and convert these areas from carbon stores to carbon sources [4]. SOC mineralization is an important process, which is part of the carbon cycle and is largely controlled by carbon quality [5]. According to Schimel and Schaeffer [6], the structure and availability of SOC greatly influence SOC mineralization.

Most soil-dwelling microorganisms are heterotrophic; they cannot synthesize their own nutrients and depend on soil organic matter as a source of energy and carbon. Since

they show aerobic development, they obtain energy from the oxidation of organic materials. The difference in organic matter decomposition can be attributed to characteristics of soil microbial communities, including structural and functional diversity [7]. According to Rousk et al. [8], the two main groups of the microbial decomposer community are bacteria and fungi, and they share the function of decomposing organic matter. Naturally, the number and diversity of microorganisms are directly affected by the environment they live in, namely soil properties [9].

These microorganism communities living in soil are considered as energy and nutrient stores apart from performing the mineralization of organic matter [10]. Understanding soil C storage in terms of soil microbiology is of great importance for the development of applications that will change soil microbial properties [6,11]. SOC quality, microbial communities, and nutrient status are thought to affect SOC mineralization [8,12,13]. According to Fontaine et al. [14], the effects of the structure and activity of the microbial community on SOC decomposition are quite strong. So far, although a number of studies have acknowledged the importance of characteristics of microbial communities such as biomass, diversity, and composition in the regulation of soil C storage [15], clear conclusions regarding the relationships between these microbial community characteristics and SOC have not been established.

Hao et al. [16] claimed that microbial biomass and composition affect SOC while Degrune et al. [17] showed that microbial community diversity was not directly related to SOC. It is stated that SOC mineralization is strongly affected by exogenous substances added to soil [18]. There are studies showing that exogenous additions can increase or decrease SOC mineralization through the activities of microorganism communities, resulting in positive or negative ignition effects [19]. As a result, it is of great importance to investigate the effect of microorganisms in the soil on SOC in terms of global carbon stock and carbon mineralization.

Among microorganisms, fungi play an important role in soil biomass and soil structure and affect biochemical and biophysical mechanisms directly and indirectly [20]. Despite this important role of fungi in the soil habitat, they can cause diseases in plants in different ways. *Rhizoctonia* spp., *Fusarium* spp., *Verticillium* spp., *Sclerotinia* spp., *Pythium* spp., and *Phytophthora* spp. cause heavy losses to many crops. They share some basic features as resting structures (microsclerotia, sclerotia, chlamydospore, or oospores) which can survive for longer periods in host plant debris, soil, or organic matter without a host. Soilborne diseases are especially difficult to control, and chemical control is not effective. Therefore, biological control was accepted as a potential control strategy [21]. *Trichoderma* spp. has effective biocontrol agents against plant pathogens [22] and is a better alternative to chemicals. The genus are the active mycoparasites which are known to produce large quantities of fungi-toxic metabolites [22–24].

Soil enzymes are closely related to soil physical, chemical, and biological properties. They play an important role in nutrient cycling especially on the recycling of soil organic matter, which has a direct effect on soil properties in soil. Therefore, soil enzyme activities are closely associated with soil organic matter. Soil enzyme activities are used as main and sensitive indicators of biological processes. They can provide fast and accurate information about small changes in microbial activity, soil organic matter, pH, and other soil properties. Since enzymes are very sensitive to the changes on their environment, soil enzymes are affected by all kinds of applications on soils [25,26].

In this study, we tested the following hypotheses: (1) it is expected that the effect of different land use, which is cultivated (wheat grown) and uncultivated (natural pasture), on soil organic carbon and enzyme activity will be different; (2) soil organic carbon and soil enzyme activities can change after inoculation with fungi.

We examined the relationships among soil organic carbon and soil enzyme activities including urease, catalase, and alkaline phosphatase in different soils used as agricultural and pasturelands with destructive soilborne fungal pathogens, namely *Verticillium dahliae*,

*Rhizoctonia solani*, and *Fusarium oxysporum* in the agricultural land and *Trichoderma* sp. as bioagent against these pathogens.

## 2. Material and Methods

### 2.1. Study Site

The soil samples were collected from the agricultural land where the wheat was grown and pastureland from Eastern Anatolia Region of Türkiye (Kars). The Eastern Anatolia Region has very cold and snowy winters, while summers are cool and rainy, with harsh continental climate characteristics. It has one of the lowest average temperatures in Türkiye. It has an annual average of 3.8 °C. The average of the highest temperature values during the year occurs in July (16.1 °C) and the average of the lowest temperature values in January (−8.8 °C) [27]. Some soil characteristics are shown in Table 1.

**Table 1.** Results of some soil properties.

| Soil | Soil Characteristics | | | |
|---|---|---|---|---|
| | **Texture** | **SOC, %** | **EC, μmhos/cm** | **pH** |
| Pastureland | Silty clay | 6.971 | 156.33 | 6.5 |
| Agricultural land | Silty clay | 4.694 | 46.33 | 6.9 |

Note. SOC: Soil organic carbon; EC: Electrical conductivity.

### 2.2. Soil Sampling

For soil physical and chemical analysis, disturbed soil samples were collected at five different points with three replicates at a depth of 0–30 cm into the soil layer for both pasture and agricultural land in 2021 in Kars.

### 2.3. Fungal Cultures and Soil Treatments

Fungi used in the study: *V. dahliae* (SOR-8), *R. solani* (S-TR-6), *F. oxysporum* (HMK2-6), and *Trichoderma* sp. isolates were supplied from the culture collection of Igdır University, Faculty of Agriculture, Department of Plant Protection and Phytopathology Laboratory. The fungi isolates were cultivated on 90 mm Petri dishes containing Potato Dextrose Agar (Oxoid, Basingstoke, Hampshire, England) at 25 °C for seven days. Then, 10 discs with mycelia (5 mm diameter) cut from the edge of a colony of these isolates were placed in a plastic zipper bag (15 cm × 10 cm) which was filled with 60 g of soil, and the soil inoculum was prepared. The control was inoculated 10 agar discs with only PDA. Three replications were used per treatment. At the same time, a total of 10 mL sterile water was added into each soil sample on day 21. The soil samples were incubated at room temperature (23 ± 2 °C) for 41 days and were evaluated in treated soils after incubation.

### 2.4. Soil Analysis

Soil texture, organic carbon, soil pH, electrical conductivity, catalase activity, urease activity, and alkaline phosphatase activity were determined in disturbed soil samples. Soil texture, organic carbon, pH, and electrical conductivity were determined according to Gee and Bauder [28], Walkley and Black [29], McLean [30], and Rhoades [31], respectively.

For determining catalase activity, 20 mL of phosphate buffer (pH 7) and 10 mL of 3% $H_2O_2$ substrate solution were added to 5 g of soil. The volume (mL) of $O_2$ released within 3 min at 23 °C was measured. Controls were tested in the same way but with the addition of 2 mL of 6.5% $NaN_3$ [32].

To determine urease activity, 0.25 mL of toluene, 0.75 mL of citrate buffer (pH 6.7), and 1 mL of 10% urea substrate solution were added to 1 g soil, and the soil samples were incubated for 3 h at 37 °C. The formation of ammonium was discovered spectrophotometrically at 578 nm [33].

To detect alkaline phosphatase activity, 4 mL phosphate buffer (pH 8.0) and 1 mL p-nitrophenyl phosphate (disodium salt hexahydrate) solution were added to the 1 g soil

sample, and the samples were incubated for 1 h at 37 °C. The formation of p-nitrophenol (p-NP) was explored spectrophotometrically at 410 nm [34].

### 2.5. Statistical Analysis

The data were analyzed using statistical software program *SPSS* (SPSS Inc., Chicago, CA, USA). The mean values of each group were tested by ANOVA (Analysis of variance) tests. The differences between each group were detected for statistical significance ($p < 0.05$), and the differences between specified groups were determined by the Duncan multiple comparison test ($p < 0.05$). Principal component analysis (PCA), redundancy analysis (RDA), and partial redundancy analysis (P-RDA) were used to determine the correlations between soil organic carbon, soil enzyme activities, pasture-agricultural land, and fungi using Xlstat.

## 3. Results and Discussion

### 3.1. Soil Organic Carbon (SOC)

Soil organic carbon is very sensitive to applications on soil [35,36]. Tillage, one of the land use changes in pasturelands, can affect soil organic carbon [37–39]. In the study, the higher values were detected in pastureland for all fungi and control. In pastureland, the highest SOC was in *V. dahliae* (7.46%), *Trichoderma* sp. (7.27%), *R. solani* (7.03%), control (6.97%), and *F. oxysporum* (6.7%), respectively. According to findings, it is clear that *F. oxysporum* decreased soil organic carbon while the other fungi increased under pastureland, but the difference was statistically insignificant except for *V. dahliae* and *Trichoderma* sp. (Table 2, Figure 1). Soil organic carbon was higher in pastureland than agricultural land. It means soil organic carbon decreased with agricultural practices (tillage). The decrease in soil organic carbon can be explained by the fact that tillage accelerates the decomposition of organic matter by breaking down soil aggregates [40]. In accordance with our study, many researchers reported that soil organic carbon decreased with tillage practices [41,42]. Additionally, Dalal et al. [43] and Gebresamuel et al. [44] determined that soil organic carbon was reduced when pastureland was converted to agricultural land.

**Table 2.** Effects of fungi and land uses types on soil organic carbon (%).

| Soil | Application | | | | |
|---|---|---|---|---|---|
| | Soilborne Pathogen | | | | Bioagent |
| | Control | *Fusarium oxysporum* | *Verticillium dahliae* | *Rhizoctonia solani* | *Trichoderma* sp. |
| Pastureland | 6.97 ± 0.2 cdA | 6.7 ± 0.13 dA | 7.46 ± 0.12 aA | 7.03 ± 0.07 bcA | 7.27 ± 0.16 abA |
| Agricultural land | 4.69 ± 0.24 aB | 4.65 ± 0.18 aB | 4.72 ± 0.12 aB | 4.37 ± 0.26 abB | 4.14 ± 0.12 bB |

Note. Lower case letters used horizontally (Control, *Fusarium oxysporum*, *Verticillium dahliae*, *Rhizoctonia solani*, and *Trichoderma* sp.) indicate significant differences; capital letters used vertically (Pasture-Agricultural) indicate significant differences according to Duncan multiple comparison test at $p < 0.05$ significance level.

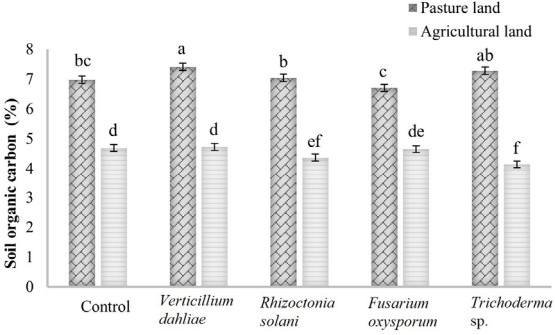

**Figure 1.** Soil organic carbon (%). Note. Different letters represent significant differences between variables according to Duncan multiple comparison test at $p < 0.05$ significance level.

The means of variables were statistically significant and the variables were categorized as "a–f" according to Duncan multiple comparison test as given in Figure 1.

The values of soil organic carbon differed for all fungi, but the difference was statistically insignificant except for *Trichoderma* sp. in the agricultural land. The values were in *V. dahliae* (4.72%), control (4.69%), *F. oxysporum* (4.65%), *R. solani* (4.37%), and *Trichoderma* sp. (4.14%), respectively (Table 2, Figure 1).

The soil organic matter sources are plant and animal (micro-macro organisms) residues. Microbial residues are important sources for soil organic matter [45–48]. Moreover, fungal residues are rich in amino acids, fungal polysaccharides, and mineral elements and act as organic materials [49]. In the study, we determined that fungi increased soil organic carbon, except *F. oxysporum* under pastureland. *Trichoderma* sp. decreased soil organic carbon, and the decrease was statistically significant, but the other fungi did not statistically affect soil organic carbon in agricultural land. The increase in pastureland may be due to fungal residues which are a source of organic matter, as stated in previous studies [45,48]. In agreement with the findings, Khan et al. [50] reported that microbial residues increased soil organic carbon. Some other researchers observed that the application of chemical fertilizers in combination with fungal residue enhanced carbon sequestration capacity of soils [51]. Additionally, Wu et al. [52] stated that the application of *Trichoderma* sp. increased the concentration of effective phosphorus, available potassium, and organic matter.

### 3.2. Soil Enzyme Activity

The type of vegetation or land use is an important factor affecting the extracellular enzyme activities of the soil. In the study, alkaline phosphatase and catalase activities were significantly affected by land use types. Phosphatase, catalase, and urease activities were measured as 117.44, 177.03, and 12.87 in the pastureland and as 109.07, 76.05, and 11.52 in the agricultural land, respectively. Similar results were also found in the treatments where the fungi were applied. However, the decrease in urease activity was statistically insignificant in control and *F. oxysporum* ($p > 0.05$); it was significant ($p < 0.05$) in *V. dahliae*, *R. solani*, and *Trichoderma* sp. (Table 3; Figures 2–4). The differences between land use types may be because of the differences in soil organic carbon [53,54]. In agreement with the findings, Tang et al. [55] and Li et al. [56] demonstrated that soil enzyme activities were higher under grassland than farmland because of soil organic carbon concentration. Błonska et al. [57] stated that woodland soils were with significantly greater enzyme activity in comparison to agricultural soils. Yang et al. [58] reported that soil enzyme activities were significantly affected by different land cover and land use types depending on the studied enzymes. Lebrun et al. [59] determined that hydrolases activities were higher in the grasslands than in the cropped soils. At the same time the fungicide applications used in agricultural areas affect the amount of soil enzymes. Chen et al. [60] indicated that dazomet applications for the control of the *F. oxysporum* on chrysanthemum significantly decreased soil catalase and urease activities.

**Table 3.** The analyzed results of soil alkaline phosphatase ($\mu$g g p-nitrophenol soil$^{-1}$ h$^{-1}$), catalase (mL O$_2$ 3 min$^{-1}$ g soil$^{-1}$), and urease activity ($\mu$g g N soil$^{-1}$ h$^{-1}$), according to applications.

| Soil | Enzyme | Application | | | | |
|---|---|---|---|---|---|---|
| | | Control | Soilborne Pathogen | | | Bioagent |
| | | | *Fusarium oxysporum* | *Verticillium dahliae* | *Rhizoctonia solani* | *Trichoderma* sp. |
| Pastureland | Alkaline phosphatase | 117.44 ± 1.21 aA | 106.09 ± 1.03 cA | 111.18 ± 2.01 bA | 110.91 ± 1.47 bA | 116.74 ± 1.10 aA |
| | Catalase | 177.03 ± 3.96 aA | 149.28 ± 8.41 cA | 90.91 ± 4.80 eA | 104.94 ± 4.89 dA | 161.29 ± 4.55 bA |
| | Urease | 12.87 ± 1.51 b | 7.67 ± 0.50 d | 12.52 ± 0.46 bcA | 10.86 ± 0.74 cA | 17.08 ± 1.30 aA |
| Agricultural land | Alkaline phosphatase | 109.07 ± 3.17 aB | 99.69 ± 0.72 bB | 87.92 ± 3.08 cB | 103.13 ± 2.92 bB | 83.89 ± 2.59 cB |
| | Catalase | 76.05 ± 5.25 aB | 65.17 ± 4.96 bB | 59.84 ± 7.19 bB | 61.21 ± 4.35 bB | 47.22 ± 2.68 cB |
| | Urease | 11.52 ± 0.69 a | 8.51 ± 1.04 b | 9.00 ± 0.80 bB | 8.13 ± 0.75 bB | 7.70 ± 0.91 bB |

Note. Lower case letters used horizontally (Control, *Fusarium oxysporum*, *Verticillium dahliae*, *Rhizoctonia solani*, and *Trichoderma* sp.) indicate significant differences; capital letters used vertically (Pasture-Agricultural) indicate significant differences according to Duncan multiple comparison test at $p < 0.05$ significance level.

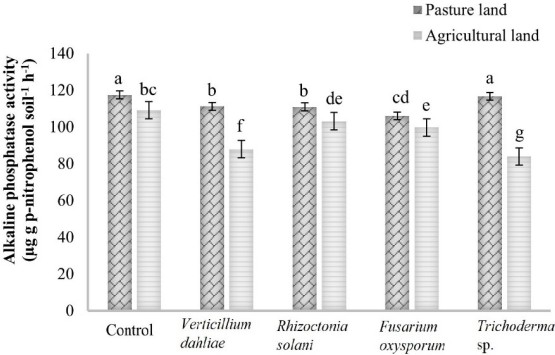

**Figure 2.** Alkaline phosphatase activity ($\mu$g g p-nitrophenol soil$^{-1}$ h$^{-1}$). Note. Different letters represent significant differences between variables according to Duncan multiple comparison test at $p < 0.05$ significance level.

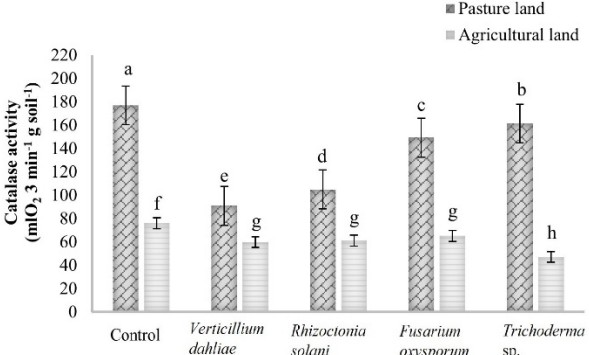

**Figure 3.** Catalase activity (mL O$_2$ 3 min$^{-1}$ g soil$^{-1}$). Note. Different letters represent significant differences between variables according to Duncan multiple comparison test at $p < 0.05$ significance level.

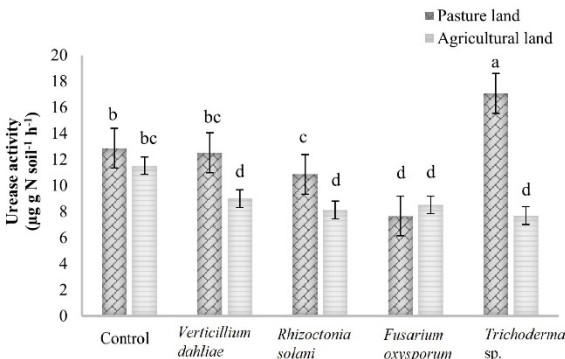

**Figure 4.** Soil urease activity ($\mu$g g N soil$^{-1}$ h$^{-1}$). Note. Different letters represent significant differences between variables according to Duncan multiple comparison test at $p < 0.05$ significance level.

The means of variables were statistically significant and the variables were categorized as "a–g" according to Duncan multiple comparison test (Figure 2).

It was seen that the means of variables were statistically significant and the variables were categorized as "a–h" according to Duncan multiple comparison test (Figure 3).

As shown in Figure 4, the means of variables were statistically significant and the variables were categorized as "a–d" according to Duncan multiple comparison test.

Another finding in the study was that alkaline phosphatase activities were significantly affected by fungi, except for *Trichoderma* sp. under pastureland ($p < 0.05$). *F. oxysporum*, *V. dahliae*, and *R. solani* decreased soil alkaline phosphatase activity. The catalase activities were significantly affected by fungi and *F. oxysporum* (149.28 mL O$_2$ 3 min$^{-1}$ g soil$^{-1}$), *V. dahliae* (90.91 mL O$_2$ 3 min$^{-1}$ g soil$^{-1}$), *R. solani* (104.94 mL O$_2$ 3 min$^{-1}$ g soil$^{-1}$), and *Trichoderma* sp. (161.29 mL O$_2$ 3 min$^{-1}$ g soil$^{-1}$) which decreased catalase activity when

compared to the control (177.03 mL $O_2$ 3 min$^{-1}$ g soil$^{-1}$). The activities of urease were significantly affected by *F. oxysporum*, *R. solani*, and *Trichoderma* sp. under pastureland ($p < 0.05$). The lowest value of urease was in *F. oxysporum* (7.67 μg g N soil$^{-1}$ h$^{-1}$), and the highest was in *Trichoderma* sp. (17.08 μg g N soil$^{-1}$ h$^{-1}$).

In agricultural land, soil enzyme activities were significantly affected by fungi ($p < 0.05$). Moreover, all fungi decreased soil alkaline phosphatase, catalase, and urease activities (Table 3; Figures 2–4). The decrease may be caused by the sampling time (the sampling time was the 41st day of inoculation). According to Christopher et al. [61], the enzyme activities increased from the seventh to the fourteenth day of sampling, and the maximum activities were observed on the fourteenth day, and then it started to decrease.

Fungi can change soil enzyme activities and microbial biomass [62,63]. *Agaricus bisporus* 100% and 150% fungal residues affected the content of soil nutrients (available N, P, and K) and soil enzymatic activities (urease, neutral-phosphatase, and catalase) at vegetative growth stage and reproductive growth stage of rice and wheat [64]. Soil enzymes such as urease, phosphatase, catalase, and saccharase are derived from various soil microorganisms. In the study, fungi decreased soil enzyme activities. In agreement with the findings, Naseby et al. [65] reported that urease activities of *Trichoderma harzianum* isolates (TH1, T4, N47, and T12) and *Trichoderma pseudokoningii* isolate (To10) showed similar trends in the control. However, *T. harzianum* (TH1, T4 and N47) significantly reduced the urease activities according to *Pythium* control. Isolates *T. harzianum* (TH1, T4, and T12) significantly decreased the alkaline phosphatase activities when compared to the *Pythium* control. Wu et al. [52] reported that urease activity and sucrase activity differed according to treatments. Some treatments significantly reduced urease activity, but some other treatments (single applications of the *Trichoderma*) caused no significant differences.

Contrary to our findings, Qin et al. [66] studied the effects of Arbuscular mycorrhizal fungi on different soil enzyme activities (protease, peroxidase, dehydrogenase, fluorescein diacetate, cellulase, β-glucosidase, trehalase, esterase, invertase chitinase, neutral phosphatase, acid phosphatase, total phosphatase, alkaline phosphatase, nitrogenase, urease, and catalase) and positive effects were determined among all enzyme activity except for polyphenol oxidase. Liu et al. [67] showed that *T. harzianum* biofertilizer significantly increased soil available sucrase and catalase activities, but the activities of urease decreased. Asghar and Kataoka [68] obtained similar results where phosphatase activity was higher with *Trichoderma* sp. than in organic materials without *Trichoderma* sp. *Trichoderma hamatum* FB10 increased enzyme activity when compared with the control, urease (12%), phosphatase (291%), catalase (27.5%), and saccharase (69%) [69]. Ji et al. [70] determined that *T. harzianum*, *Trichoderma asperellum*, *T. hamatum*, and *Trichoderma atroviride* inoculation resulted in increased soil enzyme activity including urease, phosphatase, and catalase compared to control. Xiao et al. [71] stated that *Funneliformis mosseae* increased soil nutrients and soil enzyme activities.

### 3.3. Correlation of Variables

The principal component analysis and redundancy analysis were applied for determining the correlation of variables. Principal component analysis (PCA) is a multivariate analysis method based on data reduction, taking into account the correlation between data. In the study, data were well correlated; 85.35% of the total information was explained with two PCs; and 66.28% of the information was explained by PC1 alone (Table 4).

**Table 4.** Percentage of information retained by each PC.

|  | PC1 | PC2 | PC3 | PC4 | PC5 | PC6 |
|---|---|---|---|---|---|---|
| Eigenvalue | 3.977 | 1.144 | 0.568 | 0.220 | 0.078 | 0.012 |
| Variability (%) | 66.283 | 19.070 | 9.467 | 3.672 | 1.303 | 0.206 |
| Cumulative (%) | 66.283 | 85.352 | 94.819 | 98.491 | 99.794 | 100.000 |

The amount of each of the original variables contained in the PC is defined by loading (Figure 5). By plotting the loading for two PCs, it is possible to assess the relative importance of each of the variables (i.e., fungi, SOC): the further the variable is from the origin, the more important it is. Correlations between variables are observed in the loading plot (variables with positive correlations will be placed close to each other or variables with reverse correlation will be at 180° angles to each other). The loading plot shows the importance of original variables in each PC. Soil organic carbon, urease activity, alkaline phosphatase, and catalase activities were negatively correlated with pasture-agricultural land. The figure also shows that soil organic carbon and urease activity had closer angles, which means that the variables were correlated. On the other hand, fungi and pasture-agricultural land were inversely correlated with the other variables, and pasture–agricultural land had an angle about 180° to the soil organic carbon. A 180° angle indicates that land use types were negatively correlated with soil organic carbon. Because the type of land use system is an important factor controlling soil organic carbon levels, it affects the quantity and quality of litter input, litter decomposition rates, and organic matter stabilization processes in soil. Table 5 shows that the catalase (20.89%), alkaline phosphatase (20.09%), urease (14.61%), soil organic carbon (21.98%), and pasture-agricultural land (21.26%) had a higher impact on PC1. On the other hand, fungi (1.15–82.01%) were negligible to PC1 and had a higher impact on PC2.

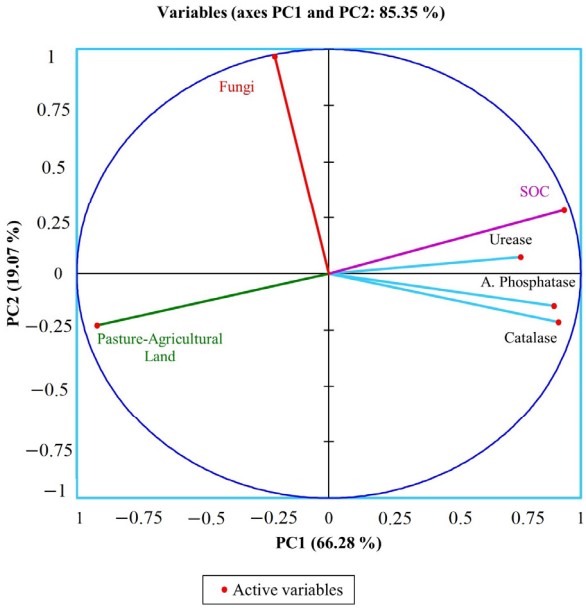

**Figure 5.** Principal component analysis (PCA) and correlation circle for principal components 1 and 2.

**Table 5.** Contribution of the variables in each PC (%).

|  | PC1 | PC2 | PC3 | PC4 | PC5 | PC6 |
|---|---|---|---|---|---|---|
| Catalase | 20.894 | 4.088 | 7.665 | 18.654 | 48.087 | 0.613 |
| Alkaline phosphatase | 20.096 | 1.792 | 5.455 | 65.215 | 7.164 | 0.279 |
| Urease | 14.611 | 0.479 | 66.434 | 15.622 | 2.024 | 0.830 |
| SOC | 21.986 | 7.008 | 4.807 | 0.048 | 14.944 | 51.207 |
| Pasture-Agricultural Land | 21.260 | 4.621 | 15.189 | 0.149 | 11.799 | 46.983 |
| Fungi | 1.154 | 82.013 | 0.449 | 0.312 | 15.983 | 0.089 |

Redundancy analysis (RDA) was used to examine the relationships among soil enzymes and fungi. Individually, the first two axes of the RDA accounted for 92.90% of the variance in the agricultural land, with the first axis accounting for 76.46% of the variance.

In pastureland, the first two axes of the RDA accounted for 98.11%, with the first axis accounting for 92.97% of the variance (Figure 6).

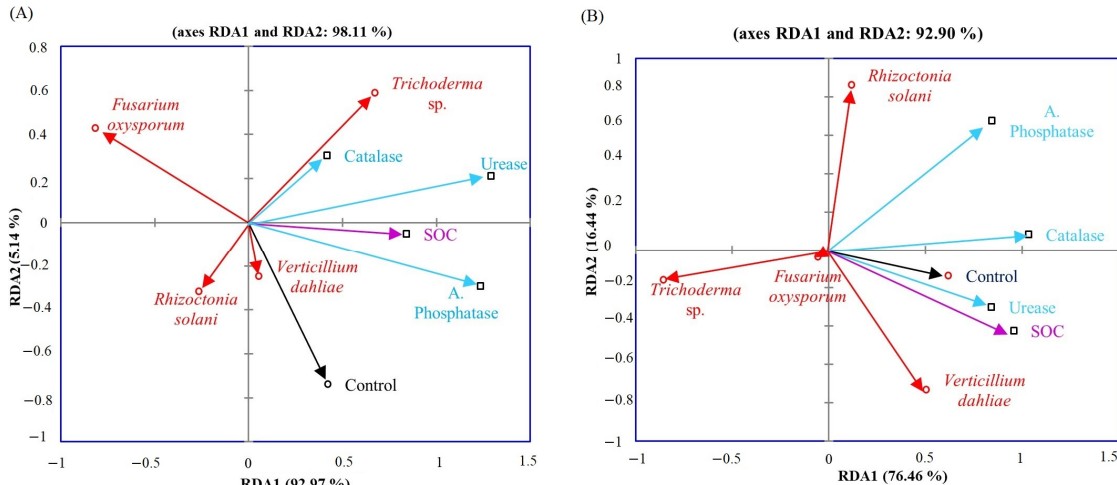

**Figure 6.** Partial-Redundancy analysis (P-RDA) of the relationship among fungi, soil enzyme activities, and soil organic carbon under (**A**) pastureland and (**B**) agricultural land.

The results of redundancy analyses for pastureland are shown in Figure 6A. According to the figure, *Trichoderma* sp. and *V. dahliae* were correlated, but *F. oxysporum* and *R. solani* were negatively correlated with the soil enzyme activities and soil organic carbon.

The agricultural land results of redundancy analyses are shown in Figure 6B. According to figure, *R. solani* and *V. dahliae* were correlated. but *Trichoderma* sp. and *F. oxysporum* were negatively correlated with the soil enzyme activities and soil organic carbon.

## 4. Conclusions

Applications of fungi (soilborne pathogens and bioagents) to the soil affected soil organic carbon and soil enzyme activities in both pasture and agricultural lands. The highest SOC and soil enzyme activities (alkaline phosphatase and catalase activities) were in pastureland. Moreover, *Trichoderma* sp. increased SOC in pastureland. Therefore, if microorganisms such as bioagents are not harmful to soil health, it is recommended that they be applied in order to increase the retention of organic carbon in soil. On the other hand, another result of this study is a decrease in soil organic carbon in the cultivated agriculture area in the same region. This has prompted us to recommend as minimum tillage as possible in order to retain carbon in soil in terms of its impact on climate change.

**Author Contributions:** Conceptualization, E.E., U.Ş. and T.G.K.; Methodology, E.E. and T.G.K.; Formal analysis, E.E. and T.G.K.; Investigation, E.E., U.Ş. and T.G.K.; Data curation, E.E. and T.G.K.; Writing—original draft preparation, E.E. and T.G.K.; Writing—review and editing, E.E., U.Ş. and T.G.K. All authors have read and agreed to the published version of the manuscript.

**Funding:** This research received no external funding.

**Institutional Review Board Statement:** Not applicable.

**Informed Consent Statement:** Not applicable.

**Data Availability Statement:** Not applicable.

**Conflicts of Interest:** The authors declare no conflict of interest.

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
