# Peer review of "Effects of Fungi on Soil Organic Carbon and Soil Enzyme Activity under Agricultural and Pasture Land of Eastern Türkiye"

_sustainability, doi:10.3390/su15031765_

Round 1
Reviewer 1 Report
The research article “Effects of fungi on soil organic carbon and soil enzyme activity under agricultural and pasture land of Eastern part of Türkiye” has been submitted by Erhan Erdel et al. The research article is very interesting. It is well written. But, it has few suggestions to improve the quality of manuscript.
1. Language editing is needed.
2. Write your research conclusion in abstract.
3. Reframe the keywords.
4. Highlight the novelty of research in abstract.
5. “Verticillium dahliae, Rhizoctonia solani and Fusarium oxysporum and Trichoderma sp. used against these pathogens”. Langauage editing is very essential.
6. Check the decimal value of pH.
7. Line 109- remove pH.
8. Table 2 and 3 Lowercase and uppercase are used. So check it.
9. Figure 1, 2,3 and 4, . All the organism names should be italics.
10. Author Contributions: and Funding: not provided.
Author Response
Dear Reviewer;
First of all, I would like to thank you for taking your valuable time for our article and helping me to improve the article by sending me your opinions and suggestions. In line with the valuable opinions and suggestions, the following corrections were made.
1-Language is edited.
2- We add our research conclusion in abstract.
3- Keywords are rewrite.
4- The novelty of research is added in abstract.
5- The sentence is corrected.
6-Value of pH is corrected.
7. Line 109- pH is removed.
8- The lower and upper cases refferes the significant differences among variables. Statistically insignificant differences are not lettered.
9- Figure 1, 2,3 and 4, . All the organism names are italics.
10. Author Contributions: and Funding are provided.
Reviewer 2 Report
L 35: This line is redundant
L 45-46: Please cite these statements
L 92: There is very less information about land use type in introduction, please elaborate more on land use type
Table 1: Please provide information about average weather conditions along with conditions for the time of study.
L 111: Shouldn’t ‘disturbed’ be ‘distributed’, or please mention what is disturbed soil sample
L 157-158: The SOC value for F. oxysporum and control had no significant difference, how did the authors conclude that it decreased the SOC.
L 173-172: The values are not significantly different, so how did the authors conclude certain values to be higher than the other.
L 179-180: This conclusion is not based on what the statistics say, please consider revising it.
Figure 1: Please consider removing the gridlines from the figure.
L 193: This conclusion is overstretched and doesn’t reflect the results. Also, there is no comparison among enzymes.
L 281: What does Pasture-agricultural land and SOC having a 180 degree angle mean: please discuss more
Author Response
Dear Reviewer;
First of all, I would like to thank you for taking your valuable time for our article and helping me to improve the article by sending me your opinions and suggestions. In line with the valuable opinions and suggestions, the following corrections were made.
1-L 35: This line is removed.
2- L 45-46: The sentences are cited as [7].
3-We have given that part a little short so that the introduction part is not too long.
The average weather conditions are written about the region in the study site section.
4- Soil samples collects as disturbed and undisturbed. Some soil analysis are made at disturbed samples and some other at undisturbed samples (bulk density).
5- Line 157-158. According to findings, it is clear that F. oxysporum decreased (The value of control is % 6.97 and F. Oxysporum is %6.7) soil organic carbon while the other fungi increased under pasture land (Table 2., Figure 1).
6-The sentence is corrected as the values of soil organic carbon differed for all fungi but the difference was satistically insignificant in the agricultural land. The values were in V. dahliae (4.72%), control (4.69%), F. oxysporum (4.65%), R. solani (4.37%) and Trichoderma sp. (4.14%), respectively. The findings suggest that all the fungi decreased soil organic carbon but V. dahliae
7-It is corrected as, in the study, we determined that fungi increased soil organic carbon except F. oxysporum under pasture land but decreased in agricultural land except V. dahliae.
8- the gridlines from the figures removed. We made some corrections.
9- We explained this part a little more.
Reviewer 3 Report
The manuscript is well-written, however, it has some scope for further improvement. I suggest some minor revisions.
Ln 309: Replace 'effected' with 'affected' Ln 311: Add an article 'the' before 'SOC' Ln 313: Replace 'to be applied' with 'that they be applied' Ln 316: Consider replacing 'minimum' with 'little' Figure legends for Fig 1-4 need to be re-written with all the details such as error bars, etc. Ln 39: The comma should be replaced with 'and the' Ln 40: Replace 'loss' with the plural form Ln 43: Replace 'influences' with 'influence' Ln 44: i.e should be followed by a comma The legend of Table 3 needs to be re-written ('Statistical analyze results' is not the correct way of writing it)
Author Response
Dear Reviewer;
First of all, I would like to thank you for taking your valuable time for our article and helping me to improve the article by sending me your opinions and suggestions. In line with the valuable opinions and suggestions, the corrections were made.
Round 2
Reviewer 2 Report
The authors concluded some major findings in the study based on faulty statistics.
'If means are not significantly different, they cannot be used for any comparisions even if actual numerical values seem to be different' - this norm is widely followed in scientific research, and conclusions are made based on that.
Author Response
Dear Reviewer;
I would like to thank you for taking your valuable time for the second time. In line with the valuable opinions and suggestions. Your statistical suggestions made important contributions to the study, thank you for drawing attention to this issue. The following corrections were made.
We did not compare the variables which were not statistically significant. As follows;
[Another finding in the study was that alkaline phosphatase activities were significantly affected by fungi except Trichoderma sp. under pasture land (p < 0.05). F. oxysporum, V. dahliae and R. solani were decreased soil alkaline phosphatase activity. The catalase activities were significantly affected by fungi and F. oxysporum (149.28 ml O2 3 min-1 g soil-1), V. dahliae (90.91 ml O2 3 min-1 g soil-1), R. solani (104.94 ml O2 3 min-1 g soil-1) and Trichoderma sp.(161.29 ml O2 3 min-1 g soil-1) decreased catalase activity when compared to the control (177.03 ml O2 3 min-1 g soil-1). The activities of urease were significantly affected by F. oxysporum, R. solani and Trichoderma sp. under pasture land. The lowest value of urease was in F. oxysporum (7.67 µg g N soil-1 h-1) and the highest was in Trichoderma sp. (17.08 µg g N soil-1 h-1).]
However, these changes did not alter the accuracy of the major findings.
Kind regards.